# Tissue-specific shaping of the TCR repertoire and antigen specificity of iNKT cells

Rebeca Jimeno[1,2], Marta Lebrusant-Fernandez[1,2], Christian Margreitter[3], Beth Lucas[4], Natacha Veerapen[5], Gavin Kelly[6], Gurdyal S Besra[5], Franca Fraternali[3], Jo Spencer[1], Graham Anderson[4], Patricia Barral[1,2]*

[1]The Peter Gorer Department of Immunobiology, King's College London, London, United Kingdom; [2]The Francis Crick Institute, London, United Kingdom; [3]Randall Centre for Cell & Molecular Biophysics, King's College London, London, United Kingdom; [4]Institute of Immunology and Immunotherapy, University of Birmingham, Birmingham, United Kingdom; [5]Institute of Microbiology and Infection, University of Birmingham, Birmingham, United Kingdom; [6]Bioinformatics and Biostatistics Science Technology Platform, The Francis Crick Institute, London, United Kingdom

**Abstract** Tissue homeostasis is critically dependent on the function of tissue-resident lymphocytes, including lipid-reactive invariant natural killer T (iNKT) cells. Yet, if and how the tissue environment shapes the antigen specificity of iNKT cells remains unknown. By analysing iNKT cells from lymphoid tissues of mice and humans we demonstrate that their T cell receptor (TCR) repertoire is highly diverse and is distinct for cells from various tissues resulting in differential lipid-antigen recognition. Within peripheral tissues iNKT cell recent thymic emigrants exhibit a different TCR repertoire than mature cells, suggesting that the iNKT population is shaped after arrival to the periphery. Consistent with this, iNKT cells from different organs show distinct basal activation, proliferation and clonal expansion. Moreover, the iNKT cell TCR repertoire changes following immunisation and is shaped by age and environmental changes. Thus, post-thymic modification of the TCR-repertoire underpins the distinct antigen specificity for iNKT cells in peripheral tissues

*For correspondence:
patricia.barral@kcl.ac.uk

Competing interests: The authors declare that no competing interests exist.

## Introduction

Most anatomical compartments, including mucosal surfaces and solid organs, host large populations of tissue-resident lymphocytes which are uniquely placed to provide local networks for immune surveillance and defence against infection (*Fan and Rudensky, 2016*). Within the families of tissue-resident lymphocytes, invariant Natural Killer T (iNKT) cells constitute the body's means to sense lipids, as antigens presented on CD1d (*Salio et al., 2014*). Accordingly, iNKT cells recognise through their T cell receptors (TCR) self-lipids as well as lipids from pathogenic bacteria, commensals, fungi or pollens; consequently contributing to anti-microbial, antitumor and autoimmune responses (*Salio et al., 2014*). Since iNKT cell activation can prevent or promote immunopathology in diverse disease contexts, the strict control of peripheral iNKT cell homeostasis is vital to regulate local immunity. Beyond the common features shared by all iNKT cells (including their CD1-restriction and innate-like properties), cells found in discrete tissues have distinct phenotypes and functions that critically modulate the outcome of immunity (*Crosby and Kronenberg, 2018*). This suggests that unique tissue-specific factors (including local lipid antigens, cytokines and/or hormones) may shape the population of iNKT cells resident in those tissues, ultimately regulating local immune responses. Consistent with this, alterations in CD1d-lipid presentation in the gut or the liver result in dysregulated homeostasis of local iNKT cells driving increased susceptibility to inflammation in these tissues (*An et al., 2014*;

*Zeissig et al., 2017*). Nonetheless, how signals from the tissue environment shape the iNKT cell population to best fit their function in their tissues of residency remains unclear.

iNKT cells have been traditionally defined by the expression of an invariant TCR α-chain (Vα14-Jα18 in mice or Vα24-Jα18 in humans) and their capacity to recognise the glycolipid antigen α-galactosylceramide (αGalCer) presented on CD1d. Despite this prototypical TCR repertoire gene usage, in recent years it has become apparent that there are variations within the iNKT cell repertoire that ultimately impact the antigen recognition capacity and consequently the functional outcomes during an immune response. In mice, although most iNKT cells express the canonical Vα14-Jα18 TCR α-chain, they can use different Vβ chains and the combination of Vβ-, Jβ-, and CDR3β-encoded residues will ultimately determine the type of ligands that iNKT cells can bind (*Cameron et al., 2015*; *Mallevaey et al., 2009*; *Matsuda et al., 2001*). Moreover, a population of αGalCer-reactive NKT cells that express Vα10 TCR and has a distinct lipid-recognition capacity has been identified (*Uldrich et al., 2011*). In humans, while the majority of αGalCer-binding iNKT cells express the prototypical Vα24Vβ11 TCR, populations of *atypical NKT cells* have been found in the blood, with cells expressing a range of TCRα and TCRβ chains that show differential recognition of lipid antigens (*Le Nours et al., 2016*; *Matulis et al., 2010*). Therefore, the so-called *invariant NKT cells* constitute a polyclonal population with a broader antigen recognition capacity than previously assumed. Since iNKT cells are tissue-resident cells an important question remains regarding whether the iNKT cell TCR repertoire (and consequently antigen specificity) is related to their anatomical location and/or shaped by the antigens that these cells encounter in peripheral tissues. Similarly, whether the iNKT cell population changes in response to environmental challenges including infection, vaccination, alterations in the diet or antibiotic use is unknown.

While the TCR repertoire is determined during thymic selection the relevance of post-thymic TCR shaping has been demonstrated for both conventional CD4$^+$ T cells and regulatory T cells (Tregs). Accordingly, the TCR repertoire of thymic and peripheral CD4$^+$ T cells (or that of recent thymic emigrants (RTE) and mature naïve T cells) are not identical, suggesting that certain clones are preferentially enriched and/or deleted in the periphery (*Correia-Neves et al., 2001*; *Houston and Fink, 2009*). Similarly, the TCR repertoire of natural Tregs is unique for individual tissues, is shaped by the local antigenic landscape and controls Treg-mediated tolerance to the tissues (*Lathrop et al., 2011*; *Lathrop et al., 2008*). In the case of iNKT cells, CCR7$^+$ iNKT cell precursors are known to emigrate from the thymus and home to peripheral tissues where they undergo further maturation (*Wang and Hogquist, 2018*). In this way peripheral signals presumably shape the iNKT cell population into a mature iNKT cell pool. However, whether they impact the antigen recognition capacity of iNKT cells in their tissues of residency remains unknown.

Here, we have investigated the impact of the anatomical location in the peripheral iNKT cell population. By combining flow-cytometry analyses, lipid-loaded tetramers and RNAseq experiments we demonstrate that the iNKT cell TCR repertoire is highly polyclonal and it is different for iNKT cells resident in various tissues resulting in differential lipid antigen recognition. In line with this, the repertoire of iNKT RTE is distinct from that of mature iNKT cells suggesting that local signals shape the mature iNKT cell population. Accordingly, the basal activation, proliferation and clonal expansion of iNKT cells is dictated by anatomical location. Moreover, the repertoire and phenotype of human iNKT cells is also different for cells found in various anatomical locations. Thus, our data uncovers a novel mechanism of tissue-specific immunoregulation that underpins the antigen-specificity of iNKT cells in different sites.

## Results

### Distinct TCR repertoire and clonal expansion for iNKT cells from various tissues

The majority of αGalCer-reactive iNKT cells express an invariant TCR α-chain (Vα14-Jα18), however the Vβ chain usage is variable, with higher percentages of cells expressing Vβ7 or Vβ8 chains. Thus, to evaluate whether the TCR repertoire of iNKT cells is related to their anatomical location we examined the expression of Vβ7 and Vβ8.1/8.2 by iNKT cells from various lymphoid tissues of WT C57BL/6 mice. We identified iNKT cells from thymus, spleen, inguinal lymph node (iLN) and mesenteric lymph node (mLN) by PBS57-loaded CD1d tetramer (CD1d-tet-PBS57; being PBS57 an analogue of

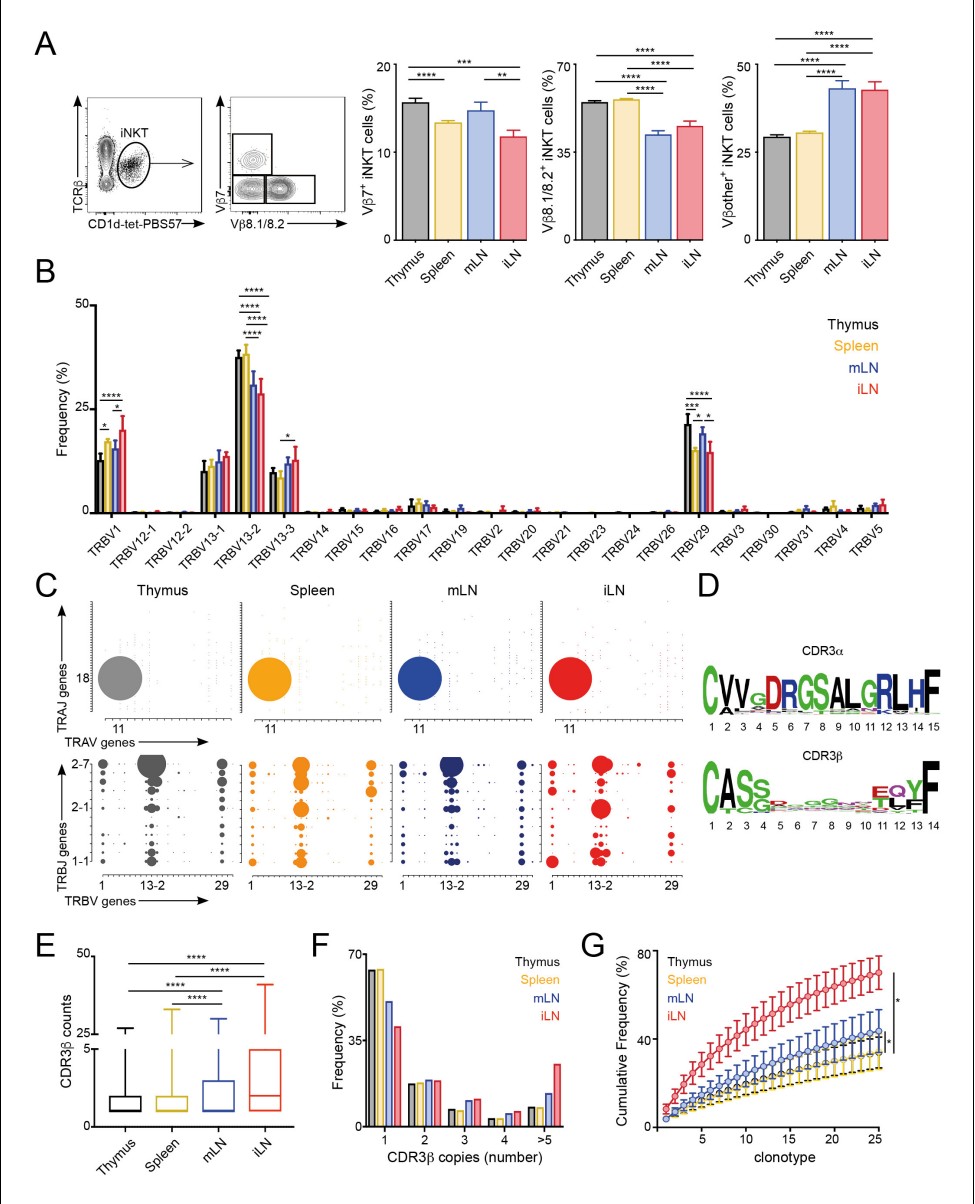

**Figure 1.** Different TCRVβ usage and clonal expansion in iNKT cells from several lymphoid tissues. (**A**) Flow cytometry plots showing gating strategy (left) and quantification (right) for iNKT cells expressing Vβ7, Vβ8.1/8.2 or Vβother (non-Vβ7 or Vβ8.1/8.2) in the depicted tissues of WT C57BL/6. Bars represent mean ± SEM. *p<0.05, **p<0.01, ***p<0.001, ****p<0.0001 paired *t*-test. n = 15 mice from five independent experiments. (**B**) Frequency of TRBV gene usage in iNKT cell TCR sequences. Frequencies are calculated from RNAseq data from four samples per tissue. Bars represent mean ± SEM. *p<0.05, ***p<0.001, ****p<0.0001, ANOVA with Tukey's multiple comparison test. (**C**) Gene usage plot (2D) showing TRAV-TRAJ (top) and TRBV-TRBJ (bottom) pairing for total TCR sequences obtained from iNKT cells isolated from the depicted tissues of WT mice. The circle size represents the percentage of sequences with each specific V-J pairing from the total TCR sequences for each tissue. (**D**) Visual representation for aa enrichment at each position for CDR3α (top) and CDR3β (bottom) sequences for iNKT cells (pooled from all tissues). Analyses were performed with sequences of 15 or 14 aa for CDR3α and CDR3β respectively. Graphics were generated with Weblogo. (**E**) Median value for counts for CDR3β sequences identified in the depicted tissues. Data obtained from RNAseq and pooled from four samples per tissue. Data have been calculated using the counts for each CDR3β sequence and expressed as box-and-whisker diagrams depicting the median ± lower quartile, upper quartile, sample minimum and maximum. ****p<0.0001 Mann-Whitney test. (**F**) Frequency of CDR3β clonotype usage in relation to the repertoire size for iNKT cells isolated from the depicted tissues (data obtained from RNAseq and pooled from four samples per tissue). Frequency of CDR3β sequences present once, twice, 3, 4 or five or more times are shown. (**G**) Cumulative frequencies occupied by the 25 most
*Figure 1 continued on next page*

*Figure 1 continued*

prevalent CDR3β clonotypes for iNKT cells isolated from the depicted tissues. Data has been calculated for sequences from four samples per tissue and represented as mean ± SEM. *p<0.05, paired *t*-test.

The online version of this article includes the following source data and figure supplement(s) for figure 1:

**Source data 1.** Different TCRVβ usage and clonal expansion in iNKT cells from several lymphoid tissues.
**Figure supplement 1.** Tissue-dependent bias for TCRVβ usage in iNKT cells.
**Figure supplement 1—source data 1.** Tissue-dependent bias for TCRVβ usage in iNKT cells.
**Figure supplement 2.** Frequency of TRBV and TRAV gene usage in iNKT cell TCR sequences.
**Figure supplement 2—source data 1.** Frequency of TRBV and TRAV gene usage in iNKT cell TCR sequences.
**Figure supplement 3.** Physicochemical properties of CDR3β sequences.
**Figure supplement 3—source data 1.** Physicochemical properties of CDR3β sequences.

αGalCer) and TCRβ co-staining (*Figure 1*; *Figure 1—figure supplement 1A*). Within the iNKT cell population we found that the percentage of Vβ usage varied according to anatomical location, and particularly iNKT cells from LNs showed decreased percentage of cells using Vβ8.1/8.2 chains and increased percentages of cells using 'other' Vβs (other than Vβ7 or Vβ8.1/8.2) in comparison with cells from thymus or spleen (*Figure 1A*). We also observed significant differences in TCRVβ usage for iNKT cells identified in non-lymphoid tissues including liver, lung and small intestinal lamina propria (SI-LP) (*Figure 1—figure supplement 1A and B*). Interestingly, we detected the same shift in TCRVβ usage after iNKT cells (enriched from spleen and thymus) were adoptively transferred into congenic mice. Twelve days after transfer, donor cells found in LNs showed a decreased frequency of TCRVβ8.1/8.2 usage in comparison with cells found in the spleen, recapitulating the TCRVβ usage of endogenous iNKT cells (*Figure 1—figure supplement 1C*). Thus, all together this data indicates that the TCRβ repertoire of peripheral iNKT cells varies according to their anatomical location.

To get an unbiased overview of the properties and TCR repertoire of peripheral iNKT cells, we sort-purified iNKT cells from various lymphoid tissues (thymus, spleen, iLN, mLN), and performed whole transcriptome RNA sequencing (RNAseq). This method, not only enabled identification of the differentially expressed genes and pathways in iNKT cells from individual tissues, but also analyses of iNKT cell clonality and TCR repertoire (*Brown et al., 2015*; *Li et al., 2016*). Thus, using our RNA-seq data we took advantage of MiXCR software to identify the CDR3 sequences, V, D and J segments and the repertoires for iNKT cells isolated from individual tissues (*Bolotin et al., 2015*; *Bolotin et al., 2013*). With this approach, we identified a total of approximately 80,000 (productive) TCR sequences (*Figure 1—figure supplement 2A*) obtained from four biological replicates per tissue. As expected, the majority of TRAV sequences (>96%) identified in all of the tissues corresponded to the canonical TRAV11-TRAJ18 rearrangement (Vα14-Jα18) and showed highly conserved CDR3α sequences (*Figure 1—figure supplement 2B*). In agreement with our flow-cytometry data TRB29 (Vβ7) and TRB13 (Vβ8) segments represented the majority of TCRβ usage in all the tissues and the percentage of usage for each gene differed between tissues (*Figure 1B–C*, *Figure 1—figure supplement 2B*). Specifically, the percentage of the most abundant TRBV13-2 gene (Vβ8.2) was decreased in samples from LNs in comparison with spleen or thymus and we also detected significant differences between tissues in the frequency of TRBV29 (Vβ7) and TRBV1 (Vβ2; *Figure 1B*). Moreover, we detected a highly diverse TRBJ usage and TRBV-TRBJ pairing with notable differences for iNKT cells from individual tissues (*Figure 1C*), which could suggest the existence of preferred TRBV-TRBJ pairing and/or the expansion (or deletion) of certain iNKT cell clones.

We found the CDR3β sequences to be highly polyclonal without conserved motives or specific residues associated with individual tissues (*Figure 1D*, *Figure 1—figure supplement 3*). Comparative analyses of length and physicochemical properties for CDR3β amino acid (aa) sequences (including hydrophobicity (based on the Kyte–Doolittle scale [*Kyte and Doolittle, 1982*]), isoelectric point (pI, according to EMBOSS [*Rice et al., 2000*]), frequency of polar (D, E, H, K, N, Q, R, S, T), aliphatic (A, I, L, V) or aromatic (F, H, W, Y) residues) did not find significant differences in sequences from iNKT cells isolated from various tissues (*Figure 1—figure supplement 3*). However, when we assessed the distribution of the iNKT cell TCR clonotype size (the proportion of CDR3β sequences which occur once, twice, etc. in a specific tissue), we detected a higher clonal size for iNKT cells from iLN and mLN in comparison with splenic and thymic iNKT cells (*Figure 1E–G*). Accordingly,

while in spleen and thymus CDR3β sequences appearing only once in our data-set represented around 65% of the total sequences, this percentage decreases to 50% in mLN and 40% in iLN (*Figure 1E–F*). We further assessed individual iNKT TCR repertoires for evidence of clonal expansion by using cumulative frequency curves to measure the 25 most prevalent clonotypes (*Figure 1G*). These analyses provided evidence for increase in clone sizes in iLN and mLN TCR repertoires in comparison with iNKT cells from spleen or thymus.

Thus, all together this data demonstrates that the TCRVβ repertoire and clonal expansion of iNKT cells varies according to their anatomical location.

## Peripheral shaping of the iNKT cell TCR repertoire

Once selected in the thymus, the TCR repertoire of conventional T cells is further shaped in peripheral lymphoid organs possibly through differential modulation of the expansion and survival of individual T cells (*Correia-Neves et al., 2001*; *Houston and Fink, 2009*). To analyse whether the TCR repertoire of iNKT cells is modulated after their arrival to peripheral organs, we compared the TCRVβ usage for iNKT cells just arriving to the tissues after thymic selection (iNKT RTE) with that of resident iNKT cells that have been in the tissues for several weeks (*Figure 2*, *Figure 2—figure supplement 1*). To identify iNKT RTE we took advantage of Rag2$^{GFP}$ reporter mice (*Boursalian et al., 2004*). In these animals GFP identifies RTE in peripheral tissues and the label is brightest in the youngest RTEs and decays over time until it can no longer be detected on T cells that have been in the periphery for more than 3 weeks (*Boursalian et al., 2004*). We identified small populations of GFP$^+$ iNKT RTE in all the analysed tissues of Rag2$^{GFP}$ reporter mice (*Figure 2A*). In the thymus the majority of iNKT cells are mature tissue-resident cells (GFP$^-$) but there is a population of GFP$^+$ iNKT cells that also includes cells in early developmental stages (*Wang and Hogquist, 2018*). In agreement with this, GFP expression was higher in thymic iNKT RTE but comparable in RTE from spleen or LNs (*Figure 2A*). We found that the TCRVβ usage was different when we compared RTE and resident (GFP$^-$) iNKT cells within the peripheral tissues of Rag2$^{GFP}$ animals (*Figure 2B*) with a consistent increase in the percentage of TCRVβ7$^+$ in the resident iNKT cell population compared to the recently arrived counterparts.

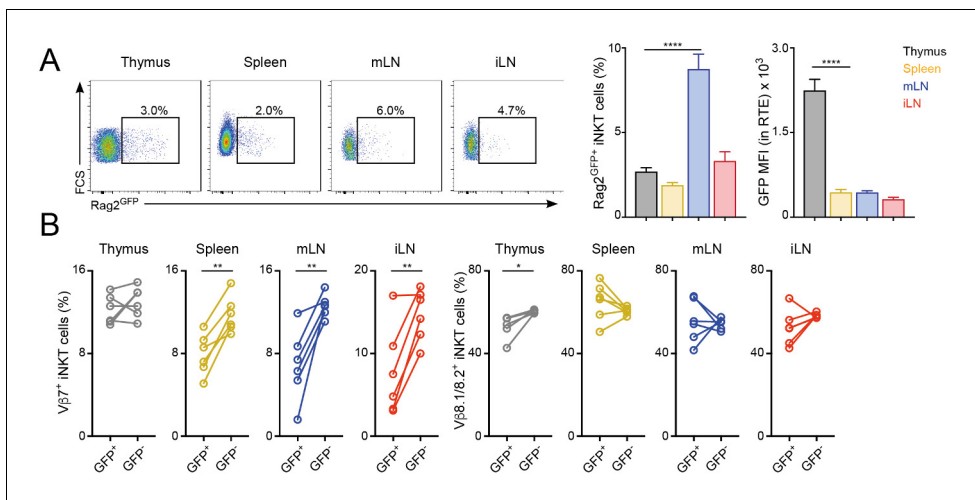

**Figure 2.** Distinct TCRVβ usage for iNKT RTE. (A–B) iNKT RTE were identified as GFP$^+$ cells in the tissues of RAG2$^{GFP}$ mice (6–9 weeks/old). (A) Flow-cytometry (left) and quantification (right) showing the percentage of RAG2$^{GFP+}$ iNKT cells and the MFI for the GFP expression in RTE in the depicted tissues. (B) Frequency of Vβ7 or Vβ8.1/8.2-expressing GFP$^+$ and GFP$^-$ iNKT cells from the tissues of RAG2$^{GFP}$ mice. Bars represent mean ± SEM. *p<0.05, **p<0.01, ****p<0.0001 two-tailed unpaired (A) or paired *t*-test (B). n = 6 mice from two independent experiments.

The online version of this article includes the following source data and figure supplement(s) for figure 2:

**Source data 1.** Distinct TCRVβ usage for iNKT RTE.

**Figure supplement 1.** TCRVβ usage for iNKT cell precursors.

**Figure supplement 1—source data 1.** TCRVβ usage for iNKT cell precursors.

To further explore the peripheral shaping of the iNKT cell repertoire, we took advantage of a recently described strategy to identify iNKT cell precursors (CCR7$^+$PD-1$^-$Qa2$^{low}$ iNKT cells) which have been shown to emigrate from the thymus to the periphery where they terminally differentiate (*Wang and Hogquist, 2018*). Using this combination of markers, we detected populations of iNKT precursors in thymus and spleen (as previously described), but also in peripheral LNs (*Figure 2—figure supplement 1A*). Notably, CCR7$^+$PD1$^-$Qa2$^{low}$ and Rag2GFP$^+$ iNKT cell populations don't fully overlap and while Rag2GFP$^+$ NKT cells are enriched for CCR7$^+$ cells, not all CCR7$^+$ NKT cells are RTE (*Figure 2—figure supplement 1B*) (*Wang and Hogquist, 2018*). Nevertheless, when we compared the TCRVβ usage for CCR7$^+$PD-1$^-$ iNKT cell precursors and 'mature' iNKT cells (excluded from the CCR7$^+$PD-1$^-$ gate) we observed consistent changes in Vβ usage with an increase in the percentage of TCRVβ7$^+$ cells in the mature iNKT cell population compared to the recently arrived counterparts (*Figure 2—figure supplement 1C*). Thus, these results suggest that after arrival to the tissues iNKT cells are exposed to local signals that shape the newly arrived iNKT cell population into a mature iNKT cell pool with a distinct TCR repertoire.

## LN iNKT cells show increased proliferation which is associated with a distinct TCRβ repertoire

To investigate the processes shaping the iNKT population in the different lymphoid tissues we studied the gene expression profile for iNKT cells isolated from thymus, spleen, mLN and iLN (*Figure 3*, *Figure 3—figure supplement 1A,B*). Transcriptome analyses showed around 200–400 genes that were differentially expressed amongst iNKT cells from various tissues (e.g. mLN and splenic iNKT cells differed by ~200 genes; adjusted *p* value < 0.01, fold change >1.5) (*Figure 3A*). Compared to spleen iNKT cells, cells from thymus, iLN and mLN showed differential expression of transcripts encoding transcriptional regulators (e.g. *Fos, Fosb, Egr1, Egr3*), nuclear factors (e.g. *Na4r1, Tcf7*), cytokine/chemokine receptors (e.g. *Ccr8, Ccr4, Il12r2b*) and molecules related to cytotoxicity (e.g. *Gzma, Klrg1*). Interestingly, iNKT cells from iLN and mLN (but not from thymus) showed upregulation of genes related to T cell activation/TCR signalling including *Na4r1* (Nur77), *Icos, Cd28, Fos, Fosb* as well as the chemokine receptors *Ccr4* and *Ccr8* which are upregulated after TCR-mediated activation (*D'Ambrosio et al., 1998*; *Sallusto et al., 1999*) (*Figure 3B*). Conversely, negative regulators of TCR signalling such as *Dok2* or *Ptpn22* were downregulated in LN cells vs iNKT cells from spleen or thymus (*Figure 3B*). Indeed, gene ontology enrichment analysis comparing the transcriptome of LN iNKT cells with cells from spleen and thymus, demonstrated significant enrichment in LN iNKT cells for genes encoding molecules related to positive regulation of T cell activation and positive regulation of cellular proliferation (*Figure 3—figure supplement 1A,B*). Hence, these results suggest that the basal activation of iNKT cells is distinct for cells found in various lymphoid tissues.

Amongst the significantly changed genes liked to T cell activation, the nuclear receptor transcription factor Nur77 (*Nr4a1*) is induced rapidly upon TCR stimulation and in a manner proportional to TCR signalling intensity (*Moran et al., 2011*) and was found to be upregulated in LN iNKT cells in comparison with thymic and splenic iNKT cells. We confirmed this result by taking advantage of Nur77$^{GFP}$ reporter mice in which we observed that iNKT cells from mLN and iLN express high levels of GFP while cells from spleen and thymus are predominantly GFP$^-$ (*Figure 3C*). Furthermore, the activation markers ICOS and CD25 and the chemokine receptor CCR8 were also found to be expressed at higher levels in LN iNKT cells in comparison with cells from spleen (*Figure 3C*). As LN iNKT cells showed an enrichment of genes involved in positive regulation of cellular proliferation we analysed iNKT cell proliferation by measuring the expression of the proliferation marker Ki-67 and the incorporation of EdU in vivo (*Figure 3D–F*). We detected a consistently higher expression of Ki-67 (*Figure 3D–E*) and higher EdU incorporation (48 hr after EdU injection; *Figure 3F*) in LN iNKT cells vs. splenic or thymic cells, confirming a higher degree of local proliferation for iNKT cells in LNs. Thus, in homeostatic conditions LN iNKT cells show increased TCR signalling, activation and proliferation in comparison to splenic or thymic cells.

To investigate whether the increased proliferation of LN iNKT cells is related to their increased clonal expansion and contributes to their distinct TCRβ repertoire we analysed the TCRVβ usage of proliferating (EdU$^+$) vs non-proliferating (EdU$^-$) iNKT cells after EdU administration in vivo (*Figure 3G*, *Figure 3—figure supplement 1C*). These experiments showed a bias in the TCRVβ repertoire, with an enrichment on Vβ7 and a decrease in Vβ8.1/8.2 usage in proliferating vs non-proliferating cells. In agreement with this data, when we compared the TRBV repertoire of sequences

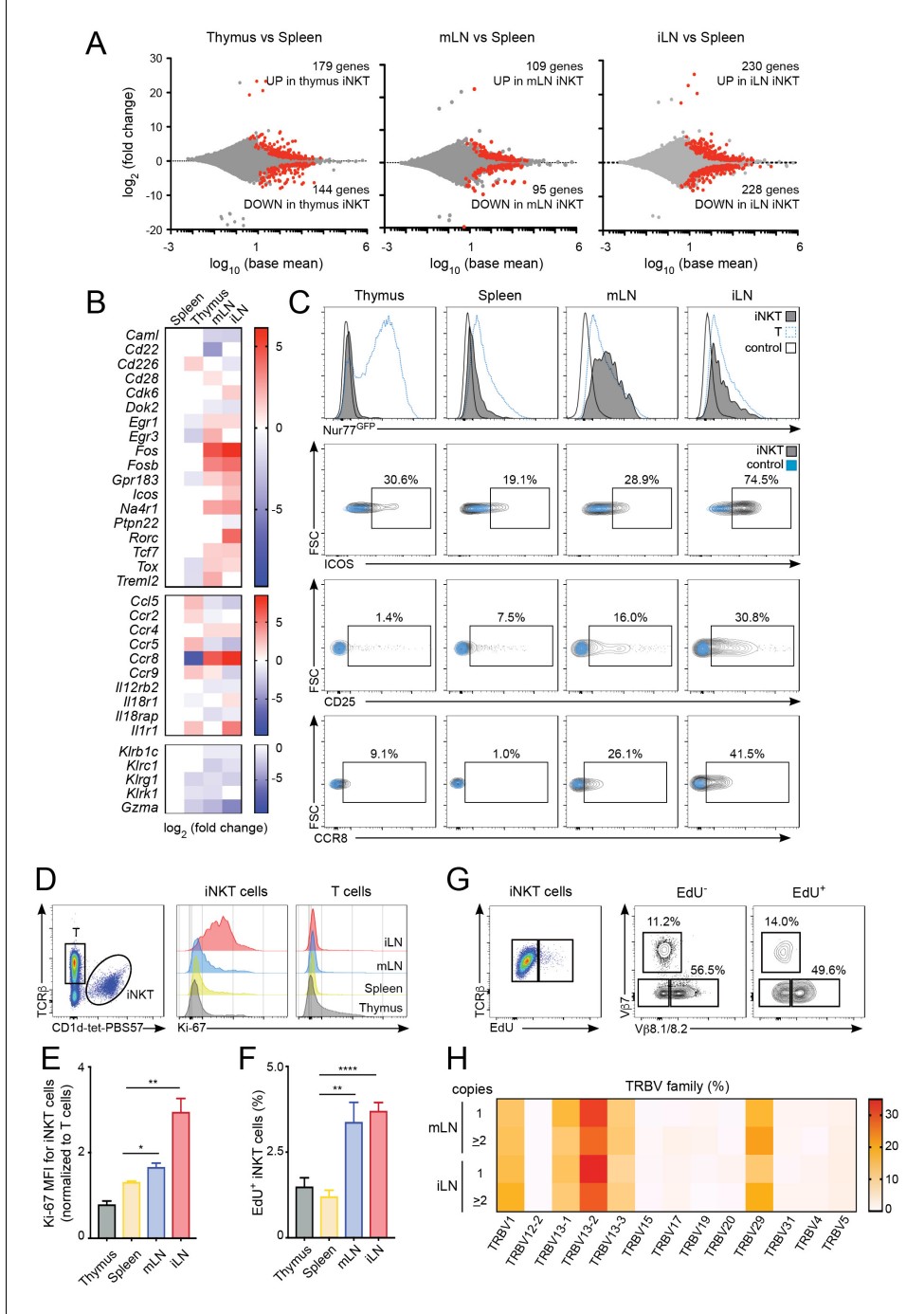

**Figure 3.** Increased activation and proliferation of LN iNKT cells. (**A**) Plots show differentially expressed genes for pairwise comparisons from iNKT cells from the depicted tissues. A fold change cut-off of 1.5 and adjusted *p*-value cut off of 0.01 were applied to colour code differentially expressed genes on the plot (red). The numbers of differentially expressed genes are indicated in the graphs. (**B**) Heat map showing RNAseq analyses of selected transcripts significantly changed in iNKT cells from thymus, mLN or iLN versus spleen. n = 4 samples. (**C**) Top, Representative flow-cytometry histograms showing GFP expression in iNKT cells (grey) and T cells (blue) from the depicted tissues of Nur77$^{GFP}$ mice. iNKT cells from the tissues of Nur77$^{GFP-}$ mice are shown as control (empty profile). Bottom, representative flow-cytometry plots showing CD25, ICOS and CCR8 expression in iNKT cells from the tissues of WT mice (grey) and control (blue). (**D–E**) Ki-67 expression in iNKT cells and T cells from the depicted tissues of WT mice. Ki-67 MFI for iNKT cells (**E**) is related to T cells from each tissue. Bars represent mean ± SEM. *p<0.05, **p<0.01, two-tailed unpaired *t*-test. n = 5 mice from two independent experiments. (**F**) Quantification of

*Figure 3 continued on next page*

*Figure 3 continued*

EdU incorporation for iNKT cells from the depicted tissues after 48 hr of EdU administration in vivo. Bars represent mean ± SEM. **p<0.01, ****p<0.0001 two-tailed unpaired *t*-test. n = 6 mice from three independent experiments. (G) Representative flow-cytometry plot showing EdU incorporation in iNKT cells (left) and frequency of Vβ7 or Vβ8.1/8.2-expressing EdU$^+$ and EdU$^-$ iNKT cells (right). n = 4 mice. (H) Heat map representation of the frequency of TRBV amongst low-abundance (one copy) or more abundant (>2 copies) CDR3β sequences obtained from iLN and mLN as indicated. Data obtained from RNAseq and pooled from four samples per tissue.

The online version of this article includes the following source data and figure supplement(s) for figure 3:

**Source data 1.** Increased activation and proliferation of LN iNKT cells.
**Figure supplement 1.** Gene expression analyses for iNKT cells from various tissues.
**Figure supplement 1—source data 1.** Gene expression analyses for iNKT cells from various tissues.

present only once in our RNAseq-derived dataset vs more abundant sequences (two copies or more), we detected increased frequency of TRBV29 (Vβ7) and decreased TRBV13-2 gene usage (Vβ8.2) in abundant vs low-copy sequences (*Figure 3H*). Thus, this data suggests that LN iNKT cells have increased basal activation and proliferation that contribute to shape the local iNKT cell TCR repertoire.

## The anatomical location governs the basal activation and TCRVβ usage of all iNKT cell subsets

iNKT cells are a heterogeneous population that can be classified into several subsets based on the expression of signature transcription factors: NKT1 (T-bet$^+$); NKT2 (PLZF$^{hi}$); NKT17 (RORγt$^+$) (*Engel et al., 2016*; *Lee et al., 2013*). Because iNKT cell subpopulations are present at different proportions in the various lymphoid organs we evaluated whether the changes in proliferation, basal activation and TCR repertoire could be due to different iNKT subsets present in those tissues. As previously reported, in C57BL/6 mice we found that NKT1 cells represent the majority of iNKT cells in thymus, spleen, mLN and iLN, but we also detected significant proportions of NKT2s and NKT17s in LNs (*Figure 4A*). Importantly, we found that the increased expression of activation and proliferation markers in LN iNKT cells was evident for all iNKT cell subpopulations (*Figure 4B*). As such, the expression of Ki-67, Nur77, ICOS and CD25 was higher in NKT1, NKT2 and NKT17 cells from LNs vs their splenic counterparts, indicating that the tissue environment controls the basal activation and proliferation of iNKT cells regardless of the subset to which they belong. In the same line, we found that the anatomical location led to variation in TCRVβ usage within individual iNKT cell subsets (*Figure 4C*). For instance, within the NKT1 population we detected a lower frequency of Vβ8.1/8.2 in cells from the LNs vs spleen or thymus. Likewise, Vβ7 usage was significantly different in all subsets when comparing cells from different tissues. In line with these results, we also detected significant differences in TCRVβ usage associated with the tissue of residency when iNKT cells were subdivided on the basis of their expression of CD4 and/or NK1.1 (*Figure 4—figure supplement 1*). Thus, all together this data indicates that the anatomical location shapes the basal activation, proliferation and TCR repertoire of all iNKT cell subsets.

## Differential lipid antigen recognition for iNKT cells in peripheral tissues

Next, we investigated the functional relevance of the distinct TCRβ repertoire for iNKT cells found in different anatomical locations. Since the TCRβ usage in iNKT cells modulates lipid antigen recognition (*Cameron et al., 2015*; *Florence et al., 2009*) we investigated the antigen binding capacity for iNKT cells in individual tissues. To do this, we costained iNKT cells from various tissues (thymus, spleen, mLN, iLN, SI-LP, liver, lung) with CD1d-tet-PBS57 and CD1d-tetramers loaded with different glycolipid antigens including the αGalCer analogue OCH, αGlucosylCeramide (C26; αGlcCer), ßGlucosylCeramide (C24; βGlcCer) and ßGalactosylCeramide (C12; βGalCer). Within the CD1d-tet-PBS57$^+$ iNKT cell population we identified distinct populations of iNKT cells binding OCH and αGlcCer in all analysed tissues (*Figure 5A–B*, *Figure 5—figure supplement 1*), but not βGlcCer or βGalCer (data not shown). Interestingly, the percentages of cells binding to the various lipid antigens was different amongst tissues (*Figure 5A–B*, *Figure 5—figure supplement 1*). For instance, while most iNKT cells bind αGlcCer and OCH in thymus, spleen, liver and lung, in iLN, mLN and SI-LP we

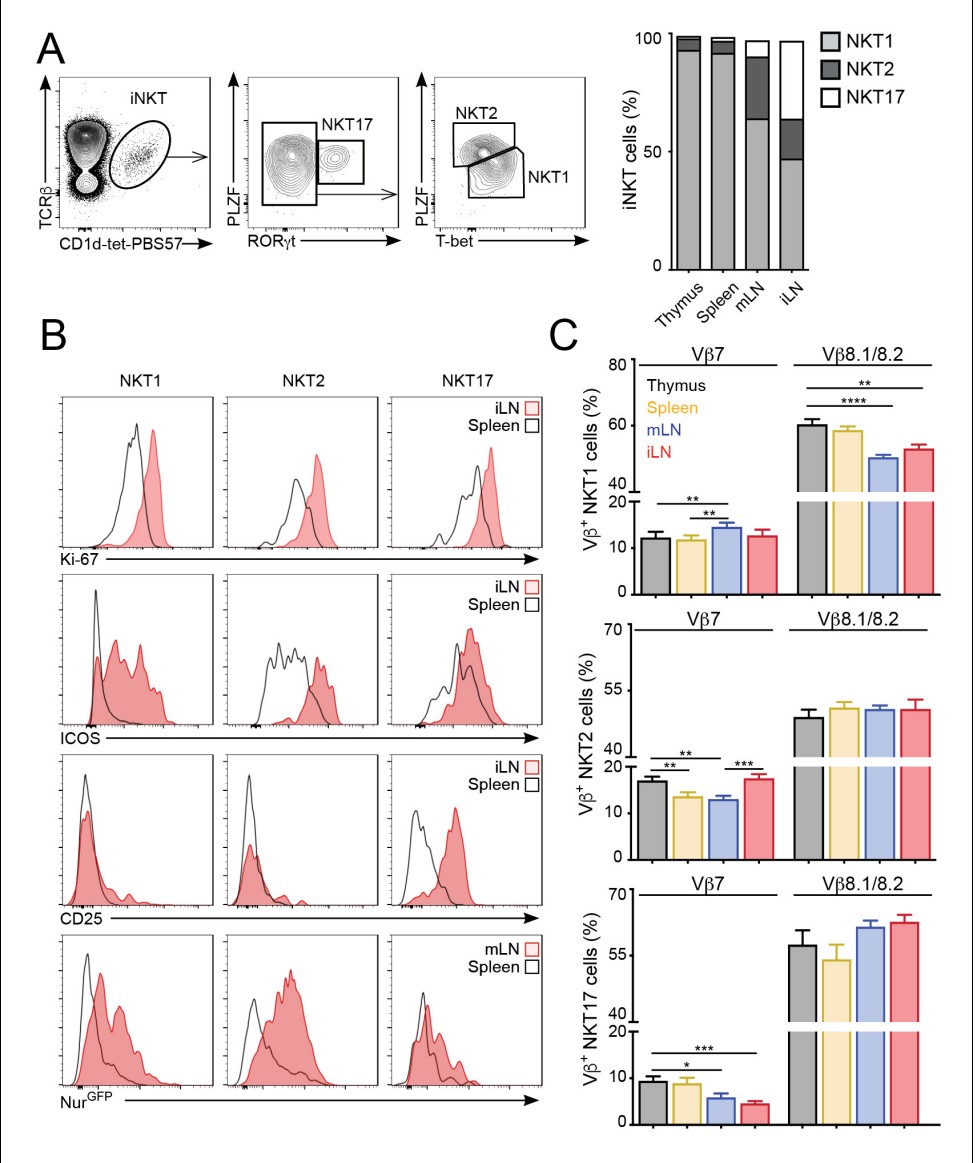

**Figure 4.** The tissue of origin dictates the basal activation and TCRβ repertoire of all iNKT subsets. (**A**) Analysis of iNKT cell populations in the tissues of WT C57BL/6 mice, showing flow-cytometry plots (A, left) and frequency (A, right) of NKT1 (RORγt⁻PLZF^loT-bet⁺), NKT2 (RORγt⁻PLZF^hiT-bet⁻) and NKT17 (PLZF^intRORγt⁺) cells. n = 10 mice from four independent experiments. (**B**) Top, Representative flow-cytometry plots showing Ki-67, CD25 and ICOS expression in NKT1, NKT2 and NKT17 cells from the depicted tissues. Subpopulations were identified as in (**A**). Bottom, GFP expression in iNKT cell subsets from the depicted tissues from Nur77^GFP mice. iNKT cell subsets were identified as described (*Engel et al., 2016*): NKT1 (CD27⁺NK1.1⁺), NKT2 (CD27⁺, NK1.1⁻, CD1d-Tet^hi, CD4⁺), NKT17 (CD27⁻, CD4⁻, CCR6⁺). (**C**) Frequency of Vβ7- or Vβ8.1/8.2-expressing iNKT cells within the NKT1 (top), NKT2 (middle) or NKT17 (bottom) populations in the depicted tissues of WT mice. n = 10 mice from four independent experiments. Bars represent mean ± SEM. *p<0.05, **p<0.01, ***p<0.001, ****p<0.0001 paired *t*-test.

The online version of this article includes the following source data and figure supplement(s) for figure 4:

**Source data 1.** The tissue of origin dictates the basal activation and TCRβ repertoire of all iNKT subsets.
**Figure supplement 1.** TCRVβ usage for iNKT cell subpopulations.
**Figure supplement 1—source data 1.** TCRVβ usage for iNKT cell subpopulations.

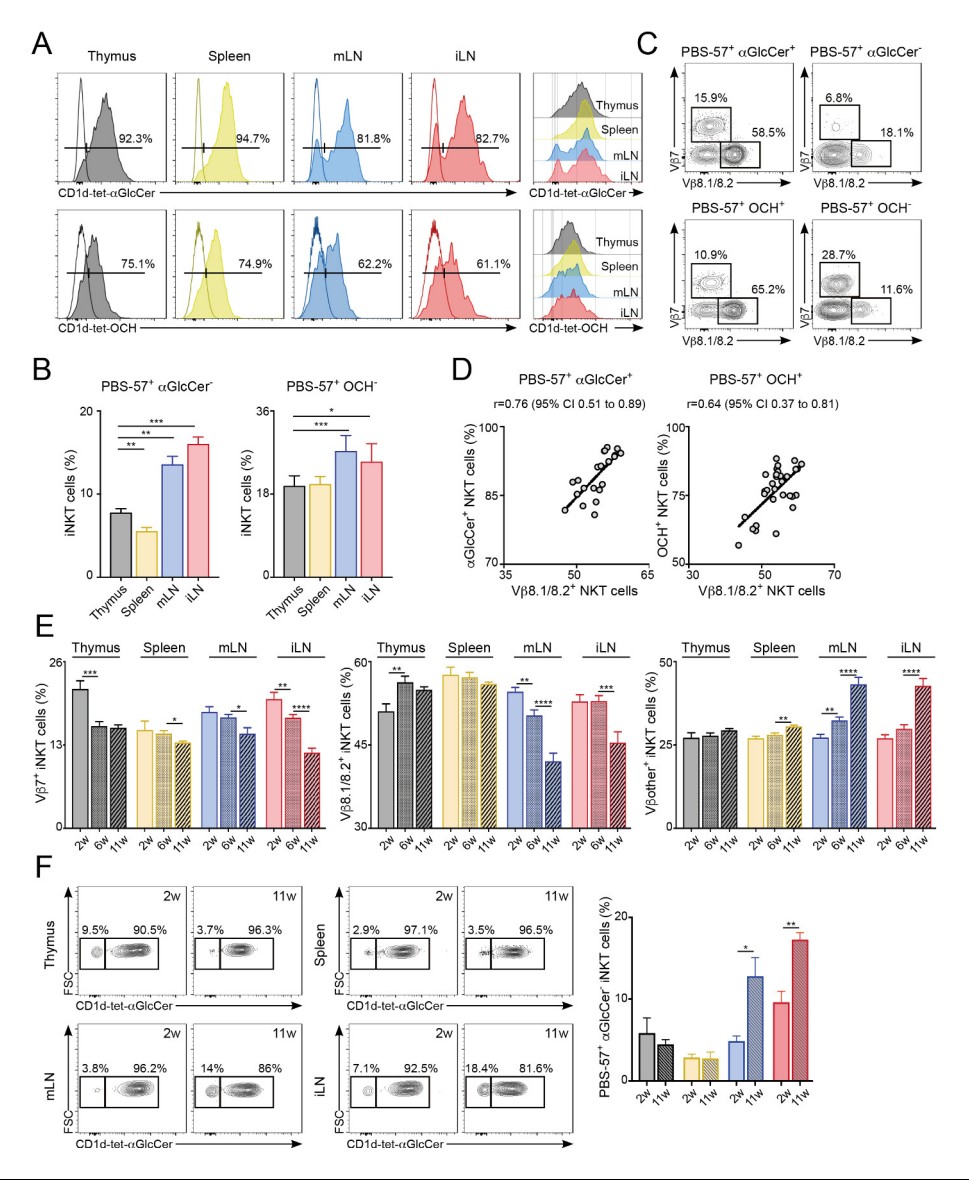

**Figure 5.** Differential lipid antigen recognition for iNKT cells from various lymphoid tissues. (A–D) iNKT cells from the depicted tissues were co-stained with CD1d-tet-PBS57 and CD1d-tet-αGlcCer or CD1d-tet-OCH. (A) Flow-cytometry profiles. (B) Quantification of αGlcCer⁻ (left) or OCH⁻ (right) iNKT cells. Bars represent mean ± SEM. *p<0.05, **p<0.01, ***p<0.001, two-tailed paired *t*-test. (C) Vβ usage for the depicted iNKT cell populations from the spleen. (D) Frequency of αGlcCer⁺ (left) or OCH⁺ (right) iNKT cells was related to the frequency of Vβ8.1/8.2 usage for each sample. Pearson correlation analyses are shown for each graph. n = 6–8 mice from 3 to 4 independent experiments. (E–F) TCR Vβ repertoire and lipid antigen recognition of iNKT cells from the depicted tissues were analysed at indicated time points (weeks of age). (E) Frequency of iNKT cells expressing Vβ7, Vβ8.1/8.2 or Vβother (no Vβ7 or Vβ8.1/8.2) in the tissues of WT C57BL/6 mice of 2, 6 or 11 weeks of age. n = 10–15 mice from 10 independent experiments. (F) Flow-cytometry profiles (left) and quantification of αGlcCer⁻ iNKT cells (right) for iNKT cells from the tissues of WT mice of 2 or 11 weeks of age as depicted. n = 3 mice from two independent experiments. Bars represent mean ± SEM. *p<0.05, **p<0.01, ***p<0.001, ****p<0.0001 two-tailed unpaired *t*-test.

The online version of this article includes the following source data and figure supplement(s) for figure 5:

**Source data 1.** Differential lipid antigen recognition for iNKT cells from various lymphoid tissues.
**Figure supplement 1.** Differential lipid antigen recognition for iNKT cells from non-lymphoid tissues .
**Figure supplement 1—source data 1.** Differential lipid antigen recognition for iNKT cells from non-lymphoid tissues.

identified clear populations of αGlcCer⁻ and OCH⁻ iNKT cells (*Figure 5A–B*, *Figure 5—figure supplement 1A–B*). Also, the TCRVβ usage shapes antigen binding as the proportion of Vβ7⁺ and Vβ8.1/8.2⁺ iNKT cells were skewed when comparing OCH⁺ vs OCH⁻ iNKT cells or αGlcCer⁺ vs αGlcCer⁻ iNKT cells (*Figure 5C*, *Figure 5—figure supplement 1C*). Consequently, we detected a strong correlation between the percentage of Vβ8⁺ iNKT cells and the binding to OCH or αGlcCer (*Figure 5D*, *Figure 5—figure supplement 1D*). Thus, this data indicates that the distinct TCRβ repertoire found in iNKT cells from various anatomical locations relates to their differential capacity for lipid antigen recognition.

It is well established that in conventional T cells there are age-dependent changes in the TCR repertoire that lead to impaired immune responses (*Yager et al., 2008*). Hence, we analysed the effect of age in iNKT cells by measuring the TCRVβ usage and lipid-binding capacity for iNKT cells in the tissues of 2, 6 and 11-week-old WT C57BL/6 mice (*Figure 5E–F*). We detected significant changes in TCRVβ usage associated to the age of the mice. For instance, the percentage of Vβ8.1/8.2⁺ iNKT cells decreases over time in iLN and mLN, whereas the percentage of cells expressing *other* Vβs in those tissues increases (*Figure 5E*). The TCR changes over-time are more evident in LNs in comparison with the spleen and the differences in TCRVβ usage for LN iNKT cells are more prominent in adult mice (6 w vs 11 w) than in younger animals (2 w vs 6 w). Importantly, the changes in the frequency of Vβ8.1/8.2⁺ iNKT cells in older mice correlate with the capacity of the cells to bind αGlcCer-loaded CD1d tetramers (*Figure 5F*). Hence, in the LNs both the frequency of Vβ8.1/8.2⁺ iNKT cells and that of αGlcCer⁺ cells decrease as mice age. Thus, this data demonstrates a differential lipid antigen recognition capacity for iNKT cells from various tissues, which is shaped by age.

## The iNKT cell TCR repertoire changes in response to immunisation and environmental challenges

We next explored the stability of the peripheral iNKT cell TCR repertoire in response to antigenic challenges and environmental changes. To analyse whether immunisation with lipid antigens induces lasting changes in the iNKT TCR repertoire we injected mice with the lipid antigen OCH and followed the changes in the iNKT cell population at 3 or 13 days after immunisation (*Figure 6A–D*). Three days after OCH administration we detected strong proliferation of iNKT cells in spleen and LNs, resulting in an increase in Ki-67 expression and in the frequency of iNKT cells in comparison with control (PBS injected) mice (*Figure 6A–C*). The vast majority (~80%) of highly proliferative iNKT cells (Ki-67ʰⁱ) expressed TCRVβ8.1/8.2. As a result, we detected a global change in the TCRVβ usage for the local iNKT cell populations in comparison with control mice with reduced frequency of TCRVβ7⁺ iNKT cells which are replaced by TCRVβ8.1/8.2⁺ cells (*Figure 6D*). Thirteen days after OCH administration Ki-67 expression and iNKT cell frequency returned to basal levels (*Figure 6B–C*). However, the TCR repertoire of the iNKT cell population was still significantly different from control mice with increased frequency of TCRVβ8.1/8.2⁺ and reduced TCRVβ7⁺ iNKT cells (*Figure 6D*). In line with these results, we also detected significant changes in the TCR repertoire of iNKT cells after immunisation with αGalCer (*Figure 6—figure supplement 1*). In response to this lipid, the frequency of TCRVβ8.1/8.2⁺ iNKT cells increased at the expense of Vβother⁺ cells while the frequency of Vβ7⁺ cells remained unchanged (*Figure 6—figure supplement 1*). Interestingly, after antigen stimulation (both in vitro and in vivo) Vβ7⁺ iNKT cells showed reduced secretion of cytokines in comparison with Vβ8.1/8.2⁺ or Vβother⁺ cells (*Figure 6—figure supplement 2*). Thus, all together this data indicates that the TCRVβ usage of iNKT cells is associated with their distinct activation and proliferation in response to antigen stimulation. As a result, exposure to lipid antigens induces a restructure in the repertoire of the iNKT cell population that is maintained even after proliferation has ceased and the population has contracted.

Commensal-derived products are known to modulate the numbers and phenotype of iNKT cells (*An et al., 2014*; *Sáez de Guinoa et al., 2018*; *Wingender et al., 2012*). Thus, we hypothesised that changes in commensal bacteria could also lead to a restructure of the iNKT cell TCR repertoire. To address this, we treated 5 week old mice with antibiotics in the drinking water for 6 weeks and analysed the TCRVβ usage in iNKT cells in those animals at 11 weeks of age (vs. control mice; *Figure 6E*). Interestingly, we found that antibiotic treatment resulted in a small but significant increase of the frequency of iNKT cells in mLN and iLN (*Figure 6—figure supplement 3*). Moreover, we detected changes in the TCRVβ usage of the iNKT cell population with a significant increase in the percentage of Vβ8.1/8.2⁺ iNKT cells in the gut-draining mLN. Added to this, we also detected

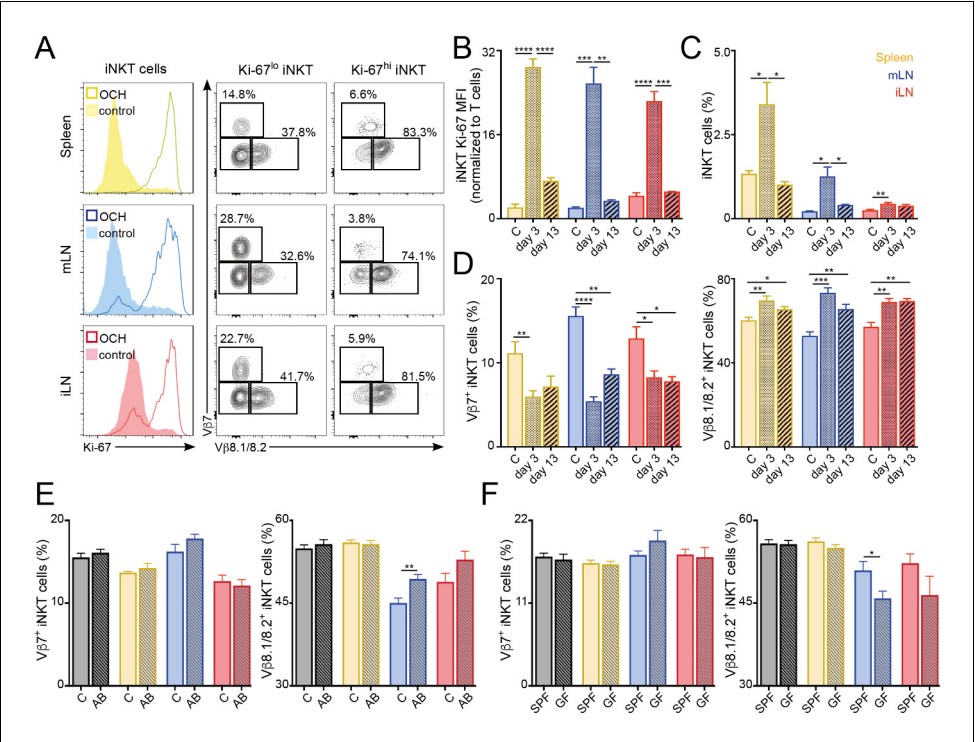

**Figure 6.** TCR repertoire of iNKT cells changes following immunisation and environmental challenges. (A–D) Mice were injected with OCH (or PBS as control, c) and iNKT cells from spleen and lymph nodes were analysed 3 or 13 days later. (A) Flow-cytometry profiles showing Ki-67 expression in all iNKT cells (left) and Vβ7, Vβ8.1/8.2 expression for iNKT cells expressing high (Ki-67hi, right) or low (Ki-67lo, middle) Ki-67 3 days after OCH administration. (B–D) Bar plots showing expression of Ki-67 (B), frequency of iNKT cells (C) and percentage of iNKT cells expressing Vβ7 or Vβ8.1/8.2 (D) at the depicted time points in spleen (yellow), mLN (blue) or iLN (red). n = 3–5 mice from 2 to 3 independent experiments. (E–F) Frequency of iNKT cells expressing Vβ7 or Vβ8.1/8.2 in the tissues of mice (Thymus = grey; Spleen = yellow; mLN = blue; iLN = red) treated with antibiotics in the drinking water vs control mice (E) or SPF vs GF mice (F). n = 10 mice (E) and n = 6 mice (F) from two independent experiments. (A–F) Bars represent mean ± SEM. *p<0.05, **p<0.01, ***p<0.001, ****p<0.0001 two-tailed unpaired t-test.

The online version of this article includes the following source data and figure supplement(s) for figure 6:

**Source data 1.** TCR repertoire of iNKT cells changes following immunisation and environmental challenges.
**Figure supplement 1.** Changes in the iNKT cell TCR repertoire following immunisation with αGalCer.
**Figure supplement 1—source data 1.** Changes in the iNKT cell TCR repertoire following immunisation with αGalCer.
**Figure supplement 2.** Cytokine secretion by iNKT cells relates to TCRVβ usage.
**Figure supplement 2—source data 1.** Cytokine secretion by iNKT cells relates to TCRVβ usage.
**Figure supplement 3.** Frequency of iNKT cells after antibiotic treatment.
**Figure supplement 3—source data 1.** Frequency of iNKT cells after antibiotic treatment.

---

changes in TCRVβ usage for mLN iNKT cells when we compared cells from the tissues of 6 week old germ-free (GF) animals with those of conventional (specific pathogen free, SPF) mice (*Figure 6F*). It is worth noting that changes in TCRVβ usage in iNKT cells were not identical in GF or antibiotic-treated mice, possibly due to the incomplete depletion of commensal bacteria and to the drastic changes in the surviving commensal populations induced by antibiotic treatment (*Hill et al., 2010*). Thus, these data suggest that modifications in the intestinal microbiota lead to changes on the TCR repertoire of iNKT cells.

## Different phenotype and TCR repertoire for iNKT cells from human tonsils and blood

Finally, we explored whether the differences in repertoire and phenotype found in murine iNKT cells resident in various tissues were also present in humans. To do this, we compared the phenotype and the TCRVα and TCRVβ usage of PBS57-binding iNKT cells from human blood and tonsils (*Figure 7*). The percentage of CD1d-tet-PBS57$^+$ iNKT cells (from total CD3$^+$ cells) in the blood was approximately 0.1%, around 10 times higher than the percentages of iNKT cells found in tonsils (*Figure 7A*). Co-staining with CD4 and CD8 revealed variable populations of cells, including CD4$^+$, CD8$^+$ and double-negative (DN) iNKT cells (*Figure 7B*). The proportion of each of these populations was variable amongst donors, but we detected a consistently higher frequency of CD4$^+$ iNKT cells within the tonsil population (mean = 72.4%) compared to blood (mean = 46.6%). Moreover, the expression of the activation markers CD25 and CD69 was also variable, but we detected an increase in the proportion of CD69$^+$ iNKT cells within the tonsils (*Figure 7C*), confirming a different phenotype for iNKT cells found in tonsils vs. blood.

While the majority of human iNKT cells express the prototypical Vα24Vβ11 TCR, populations of *atypical NKT cells* have been found in human blood, representing up to 10% of the iNKT cell population (*Le Nours et al., 2016*). These cells retain the capacity to bind CD1d-tet-PBS57 but express a range of TCRα and β chains that result in differential recognition of lipid antigens. Thus, we analysed the usage of Vα24 and Vβ11 within the iNKT cell populations from tonsils and blood (*Figure 7D–E*). As previously described, we found that the majority of CD1d-tet-PBS57$^+$Vα24$^+$ iNKT cells express TCRVβ11 in both blood and tonsils. Similarly, the majority of CD1d-tet-PBS57$^+$Vβ11$^+$ iNKT cells expressed Vα24. However, we found a variable proportion (0–10%) of iNKT cells lacking expression of Vα24 or Vβ11 in blood and tonsils. These *atypical iNKT cell* populations were found at significantly higher frequency in the tonsils in comparison to blood. For instance, Vα24$^-$Vβ11$^+$ iNKT cells represent up to 10% of the iNKT cell population in tonsils while they comprised between 0% and 3% of

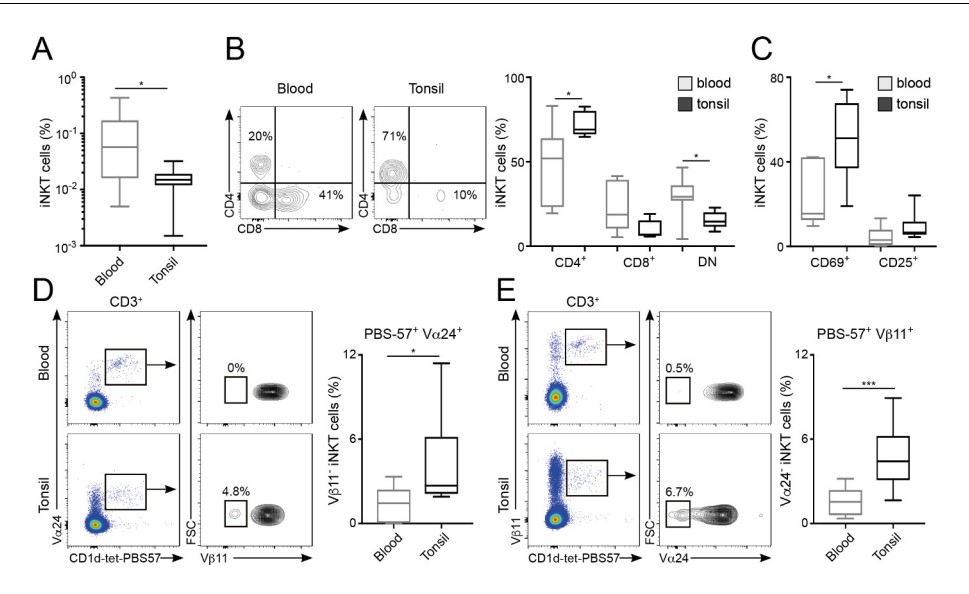

**Figure 7.** Distinct phenotype and TCR repertoire for human iNKT cells from different anatomical locations. (**A**) Mean percentage of iNKT cells (CD1d-tet-PBS57$^+$CD3$^+$) from total CD3$^+$B220$^-$CD14$^-$ cells in blood or tonsils. (**B–C**) Flow-cytometry (left) and quantification (right) showing the percentage of CD4$^+$, CD8$^+$ and DN cells (**B**) or CD69$^+$ and CD25$^+$ cells (**C**) within the iNKT cell population in the depicted tissues. (**D–E**) Flow-cytometry (left) and quantification (right) showing the percentage of Vβ11$^-$ iNKT cells within CD1d-tet-PBS57$^+$Vα24$^+$ cells (**D**); or the percentage of Vα24$^-$ cells within CD1d-tet-PBS57$^+$Vβ11$^+$ cells (**E**). (**A–E**) Data are expressed as box-and-whisker diagrams depicting the median ± lower quartile, upper quartile, sample minimum and maximum. *p<0.05, ***p<0.001, two-tailed unpaired *t*-test. n = 10 samples per tissue from four independent experiments.
The online version of this article includes the following source data for figure 7:

**Source data 1.** Distinct phenotype and TCR repertoire for human iNKT cells from different anatomical locations.

iNKT cells in blood (*Figure 7D–E*). Thus, all together this data demonstrates that the TCR repertoire and phenotype of human iNKT cells are distinct for cells found in different anatomical sites.

## Discussion

Tissue-resident iNKT cells are known to have unique properties and functions related to the tissues in which they reside, yet the antigen specificity of these populations in individual tissues remains unknown. Here, we found that the basal activation, proliferation, TCR repertoire and antigen specificity of peripheral iNKT cells are modulated by their anatomical location. While the TCR of αGalCer-reactive NKT cells has been described as 'invariant', recent studies have identified variability in TCR Vα and Vβ chains in both the mouse and human repertoires resulting in differential capacity for lipid antigen recognition (*Cameron et al., 2015*; *Le Nours et al., 2016*; *Matsuda et al., 2001*). In agreement with this data, we found that the Vβ and Jβ usage, Vβ-Jβ pairing and CDR3β in iNKT cells are highly variable and relate to their capacity for lipid recognition. Thus, the so-called *invariant* NKT cell population expresses a variable TCRVβ repertoire that differs according to their anatomical location and results in differential antigen recognition capacity for iNKT cells resident in individual tissues. Importantly, our data confirms that anatomical differences also apply to human iNKT cells. Thus, we found increased frequencies of *atypical* iNKT cells (Vα24$^-$ or Vβ11$^-$) in tonsils vs blood while the frequency of CD4$^+$ iNKT cells and CD69$^+$ iNKT cells was also different in those anatomical locations. This data is particularly relevant, as the vast majority of studies related to human iNKT cells are focused on cells isolated from the blood that may not fully recapitulate the phenotype and specificity of iNKT cells residing in peripheral tissues.

After selection in the thymus, peripheral CD4$^+$ T cell survival requires the expression of MHC. In the case of iNKT cells, while thymic development is CD1d-dependent, CD1d expression and the iNKT TCR have been proposed to be dispensable for the survival of peripheral iNKT cells (*Matsuda et al., 2002*; *Vahl et al., 2013*). However, CD1d-TCR signals are required for post-thymic maturation of iNKT cells in the periphery (e.g. NK1.1$^-$ to NK1.1$^+$ transition) (*McNab et al., 2005*) and CD1d expression on hepatocytes and CD11c$^+$ cells regulates the phenotype and numbers of iNKT cells in liver and gut respectively (*Sáez de Guinoa et al., 2018*; *Zeissig et al., 2017*). Thus, this suggests that while CD1d-TCR signals might not be required for iNKT cell survival, they can shape the peripheral iNKT cell population. Our data supports a model by which after selection in the thymus, the TCR repertoire of the iNKT cell population is subjected to further shaping in the periphery resulting in differential capacity for lipid recognition by iNKT cells resident in individual tissues. Another (but not exclusive) explanation for the tissue-skewed TCR repertoire could be that after thymus export the TCR of individual iNKT cells is linked to homing in specific tissues. However, it has been shown that the iNKT cell TCR specificity doesn't significantly affect iNKT cell homing in the tissues. Accordingly, iNKT cells expressing a variety of TCRVβ chains and CDR3β sequences generated in retrogenic or transnuclear mice were found to home efficiently in a variety of tissues irrespectively of their TCR specificity or Vβ usage (*Clancy-Thompson et al., 2017*; *Cruz Tleugabulova et al., 2016*). In line with this, our own analyses of the iNKT cell CDR3β sequences did not reveal any obvious correlations between specific residues or their physicochemical properties and their tissue of origin. Thus, while our data doesn't preclude that after thymic egress certain iNKT cell clones may preferentially home in specific tissues, it supports a model in which local signals shape the TCR repertoire and specificity of the iNKT cell population after their arrival to the tissues.

Tissue-specific programming has been described for various tissue-resident immune cell populations including macrophages, innate lymphoid cells or Tregs whose properties are controlled by local tissue-derived signals (*Miragaia et al., 2019*; *Okabe and Medzhitov, 2014*). In the case of iNKT cells, previous studies have shown that while cells from spleen and liver display relatively similar transcriptional programs, adipose tissue iNKT cells present a distinct transcriptional profile associated with a unique (PLZF-independent) developmental pathway (*Lynch et al., 2015*). Our data shows a relatively similar transcriptome for iNKT cells isolated from thymus, spleen and LNs with only around 200–400 genes differentially expressed in these tissues (in contrast to the thousands of genes differentially expressed by adipose vs. splenic iNKT cells *Lynch et al., 2015*). This suggests that while iNKT cells found in various lymphoid tissues likely share the same developmental program, local signals in the tissues in which they reside shape their phenotype and functions. Accordingly, iNKT cells isolated from LNs show differential expression of genes related to T cell activation, TCR signalling

and proliferation indicating that in the LNs iNKT cells are recognising lipid antigens (e.g. endogenous or from commensals) that are not *seen* by iNKT cells residing in spleen or thymus in homeostatic conditions. Thus, it is feasible to speculate that the catalogue of (endogenous and exogenous) lipids presented by CD1d in specific tissues could contribute to shape the population of iNKT cells resident in those tissues. Added to this, it is likely that other signals (such as cytokines or hormones) also contribute to modulate the phenotype and function of tissue-resident iNKT cells (*Holzapfel et al., 2014*; *Matsuda et al., 2002*).

The observation that the iNKT cell TCR repertoire is shaped by anatomical location resembles the unique repertoire observed for natural Tregs which is also shaped by environmental antigens possibly controlling Treg-mediated tolerance to the specific tissue environment (*Lathrop et al., 2011*; *Lathrop et al., 2008*). The tight control of CD1d-dependent lipid presentation and peripheral iNKT cell homeostasis are also critical to prevent local inflammation. Accordingly, dysregulation of intestinal iNKT cell homeostasis as a consequence of alteration in commensal lipids results in increased susceptibility to intestinal inflammation (*An et al., 2014*; *Olszak et al., 2012*; *Wingender et al., 2012*). Also, CD1d-lipid presentation by hepatocytes controls peripheral induction of iNKT cell tolerance in the liver, protecting from hepatic inflammation (*Zeissig et al., 2017*). Importantly, changes in the iNKT cell TCR repertoire have been also associated with autoimmune diseases as is the case in patients with diabetes (*Tocheva et al., 2017*) and rheumatoid arthritis (*Mansour et al., 2015*) and transgenic mice over-expressing an autoreactive NKT cell-TCR develop spontaneous colitis (*Liao et al., 2012*). Thus, the local tissue-specific regulation of iNKT cell immunity may have important implications for the development and progression of autoimmune and inflammatory processes.

In summary, our study demonstrates that local signals shape the populations of tissue-resident lymphocytes and suggests that exposure to different immunisations, microbial infections or environmental changes can impact and shape the host's iNKT cell TCR repertoire. These findings may inform future development of novel therapies based on the manipulation of iNKT cells for vaccination or immunotherapy.

# Materials and methods

**Key resources table**

| Reagent type (species) or resource | Designation | Source or reference | Identifiers | Additional information |
|---|---|---|---|---|
| Strain, strain background (*Mus musculus*) | Rag2-GFP: Tg(Rag2-EGFP)1Mnz | PMID: 10458165 | MGI:3784416 | |
| Strain, strain background (*Mus musculus*) | Nur77-GFP: C57BL/6-Tg(Nr4a1-EGFP/cre)820Khog/J | PMID: 21606508 | MGI:5007644 | |
| Strain, strain background (*Mus musculus*) | CD1d-KO: Del (3Cd1d2-Cd1d1)1SbpJ | PMID: 14632651 | MGI:5582477 | |
| Antibody | PBS57-loaded CD1d-tetramer (mouse) | NIH Tetramer Core Facility | https://tetramer.yerkes.emory.edu | (1:1000) |
| Antibody | anti-mouse B220 (rat monoclonal) | BioLegend | 103224 | (1:200) |
| Antibody | anti-mouse CD8α (rat monoclonal) | BioLegend | 100714 | (1:200) |
| Antibody | anti-mouse CD11b (rat monoclonal) | BioLegend | 101226 | (1:200) |
| Antibody | anti-mouse CD11c (armenian hamster monoclonal) | BioLegend | 117323 | (1:200) |
| Antibody | anti-mouse PLZF (armenian hamster monoclonal) | BioLegend | 145807 | (1:200) |

*Continued on next page*

*Continued*

| Reagent type (species) or resource | Designation | Source or reference | Identifiers | Additional information |
|---|---|---|---|---|
| Antibody | anti-mouse RORγt (mouse monoclonal) | BD Biosciences | 564722 | (1:200) |
| Antibody | anti-mouse T-bet (mouse monoclonal) | BioLegend | 644823 | (1:200) |
| Antibody | anti-mouse TCRβ (armenian hamster monoclonal) | BioLegend | 109233 | (1:200) |
| Antibody | anti-mouse Vβ7 (rat monoclonal) | BioLegend | 118306 | (1:200) |
| Antibody | anti-mouse Vβ8.1/8.2 (rat monoclonal) | eBiosciences | 46-5813-80 | (1:200) |
| Antibody | anti-mouse CCR7 (rat monoclonal) | BioLegend | 120105 | (1:100) |
| Antibody | anti-mouse PD1 (rat monoclonal) | BioLegend | 135219 | (1:200) |
| Antibody | anti-mouse Qa2 (mouse monoclonal) | BD Biosciences | 743312 | (1:200) |
| Antibody | anti-mouse CD25 (rat monoclonal) | BioLegend | 102015 | (1:200) |
| Antibody | anti-mouse Ki-67 (rat monoclonal) | BioLegend | 652425 | (1:200) |
| Antibody | anti-mouse ICOS (rat monoclonal) | BioLegend | 117405 | (1:200) |
| Antibody | anti-mouse CD4 (rat monoclonal) | BioLegend | 100433 | (1:200) |
| Antibody | anti-mouse NK1.1 (mouse monoclonal) | eBiosciences | 11-5941-85 | (1:200) |
| Antibody | anti-mouse CD27 (armenian hamster monoclonal) | BioLegend | 124215 | (1:200) |
| Antibody | anti-mouse CCR6 (armenian hamster monoclonal) | BioLegend | 129809 | (1:200) |
| Antibody | anti-mouse CD45.1 (mouse monoclonal) | BioLegend | 110731 | (1:200) |
| Antibody | anti-mouse CD45.2 (mouse monoclonal) | BioLegend | 109805 | (1:200) |
| Antibody | anti-mouse CCR8 (rat monoclonal) | BioLegend | 150320 | (1:200) |
| Antibody | anti-mouse IL-4 (rat monoclonal) | BioLegend | 504111 | (1:200) |
| Antibody | anti-mouse IFN-γ (rat monoclonal) | BioLegend | 505810 | (1:200) |
| Antibody | PBS57-loaded CD1d -tetramer (human) | NIH Tetramer Core Facility | https://tetramer. yerkes.emory.edu | (1:500) |
| Antibody | anti-human CD3 (mouse monoclonal) | BioLegend | 300418 | (1:200) |
| Antibody | anti-human CD4 (rat monoclonal) | BioLegend | 357415 | (1:200) |
| Antibody | anti-human CD8a (mouse monoclonal) | BioLegend | 300913 | (1:200) |
| Antibody | anti-human CD25 (mouse monoclonal) | BioLegend | 302613 | (1:200) |

*Continued on next page*

*Continued*

| Reagent type (species) or resource | Designation | Source or reference | Identifiers | Additional information |
|---|---|---|---|---|
| Antibody | anti-human CD69 (mouse monoclonal) | BioLegend | 310921 | (1:200) |
| Antibody | anti-human Vα24 (mouse monoclonal) | BioLegend | 360003 | (1:200) |
| Antibody | anti-human Vβ11 (human monoclonal) | Miltenyi Biotech | 130-108-799 | (1:200) |
| Antibody | anti-human CD19 (mouse monoclonal) | BioLegend | 302223 | (1:200) |
| Antibody | anti-human CD14 (mouse monoclonal) | BioLegend | 325615 | (1:200) |
| Commercial assay or kit | Zombie (fixable viability dye) | BioLegend | 423105 | |
| Commercial assay or kit | Dynabeads Biotin binder | Invitrogen | 11047 | |
| Commercial assay or kit | Click-iT Plus EdU Flow-cytometry assay kit | Invitrogen | C10418 | |
| Chemical compound, drug | αGalCer (α-Galactosylceramide, KRN7000) | Enzo Life Sciences | ALX-306–027 | |
| Chemical compound, drug | OCH | Enzo Life Sciences | ALX-306–029 | |
| Software, algorithm | MiXCR | *Bolotin et al. (2013)* | https://mixcr.readthedocs.io/en/master/ | |
| Software, algorithm | Weblogo | *Crooks et al. (2004)* | https://weblogo.berkeley.edu | |
| Software, algorithm | Brepertoire | *Margreitter et al. (2018)* | http://mabra.biomed.kcl.ac.uk/BRepertoire_5/? | |

## Mice

CD1d-KO (on C57BL/6 background), WT C57BL/6, congenic CD45.1 WT C57BL/6 and Nur77^GFP mice were bred under specific pathogen-free (SPF) conditions at the Francis Crick Institute. Nur77^GFP mice were provided by Adrian Hayday (Francis Crick Institute). Rag2^GFP mice were bred at the University of Birmingham. Tissues from germ-free mice were obtained from the Welcome Trust Sanger Institute (Cambridge, UK). All animal experiments were approved by the Francis Crick Institute and the King's College London's Animal Welfare and Ethical Review Body and the United Kingdom Home Office.

## Human tissues

Human tissues used in this study were collected with ethical approval from UK Research Ethics Committees administered through the Integrated Research Application System. All samples were collected with informed consent. Mononuclear cells from peripheral blood and tonsils were isolated as previously described (*Zhao et al., 2018*) and cryopreserved before use.

## Murine tissue preparation

Spleen, thymus and liver were harvested and smashed through a 45 µm strainer to obtain single-cell suspensions before staining for flow-cytometry. LNs and lung were harvested and briefly digested (15 min at 37°C) with collagenase (1.5 mg/ml), DNAse (100 µg/ml) and NADase (6 mg/ml) before filtering through a 45 µm strainer. SI-LP was processed as described (*Saez de Guinoa et al., 2017*). Briefly, small intestine (excluding Peyer's patches) was flushed with cold PBS, opened longitudinally and incubated for 20 min at 37°C in HBSS 1 mM EDTA, 5% FCS. The supernatant containing epithelial cells and intraepithelial lymphocytes was discarded and the remaining tissue was incubated for

45 min at 37°C with collagenase and DNAse as above and filtered through 45 µm strainer. Cells from all the tissues were resuspended in FACS buffer (PBS 1% BSA, 1% FCS) for flow-cytometry staining.

## Flow-cytometry

Flow cytometry staining of mouse and human samples were performed in FACS buffer using the following antibodies from Biolegend or eBioscience unless specified otherwise: Anti-mouse antibodies: CD45R/B220 (RA 3-6B2), CD8α (56–6.7), CD11b (M1/70), CD11c (N418), PLZF (9E12), RORγt (Q31-378 BD Biosciences), T-bet (4B10), TCRβ (H57-587), Vβ7 (TR310), Vβ8.1/8.2 (KJ16), PD-1 (29F.1A12), CCR7 (4B12), Qa2 (1-1-2, BD Biosciences), CD25 (PC-61), Ki-67 (16A8), CD45.1 (A20), CD45.2 (104), ICOS (7E.17G9), CD4 (GK1.5), NK1.1 (PK136), CD27 (LG.3A10), CCR6 (29–2 L17), CCR8 (SA214G2), IL-4 (11B11), IFN-γ (XMG1.2). Anti-human antibodies: CD3 (HIT3a), CD4 (A161A1), CD8α (HIT8a), CD25 (BC96), CD69 (FN50), Vα24 (C15/TCR Vα24), Vβ11 (REA559, Miltenyi Biotech), CD19 (HIB19), CD14 (HCD14). PBS57-loaded mouse and human CD1d tetramers were provided by the NIH Tetramer Core Facility. Incubations were performed on ice except for CCR7 staining in which cells were incubated with antibodies for 45 min at 37°C. For transcription factor staining, cells were fixed and permeabilised with Foxp3/Transcription Factor Staining Buffer Set (eBioscience). For intracellular cytokine staining, cells were fixed and permeabilised with Fixation/Permeabilization Solution Kit (BD Biosciences). Dead cells were detected with Zombie fixable viability kit (Biolegend). Flow-cytometry data were collected on a Fortessa or Fortessa X20 flow cytometers (both from BD Biosciences) and were analysed with FlowJo software (TreeStar).

## Administration of antibiotics and EdU incorporation

Mice were orally administrated a cocktail of antibiotics: 1 mg/ml Ampicillin, 1 mg/ml Gentamicin, 1 mg/ml Neomycin, 1 mg/ml Metronidazole and 0.5 mg/ml Vancomycin (all from Sigma-Aldrich) in filtered drinking water for 6 weeks (*Jimeno et al., 2018*).

For EdU incorporation experiments mice were injected intraperitoneally with 500 µg of EdU for two consecutive days and EdU incorporation was detected with Click-iT Plus EdU Flow-cytometry assay kit (Life Technologies).

## Lipids and tetramer loading

CD1d tetramers were provided by the NIH Tetramer Core Facility. αGalCer and OCH were obtained from Enzo Life sciences. ßGalactosylCeramide (C12; βGalCer) was from Avanti Polar Lipids. αGlucosylCeramide (C26; αGlcCer) and ßGlucosylCeramide (C14; βGlcCer) were produced in house (University of Birmingham). Lipids were dissolved in 0.5% v/v Tyloxapol (Sigma) and loaded into CD1d at a three to six-fold molar excess.

## Stimulation with lipid antigens and adoptive transfer

For in vivo experiments mice were injected intravenously with 5 µg of OCH or 1 µg of αGalCer and sacrificed at the indicated time-points.

For in vitro stimulation experiments, single cell suspensions from the spleen were prepared as described above. Cells were cultured in complete media (IMDM, 10% FCS) and stimulated for 2.5 hr in the presence of 5 µg/mL αGalCer at 37°C. Brefeldin A (Biolegend) was added for the last 2 hr of the incubation period. Cells were stained for detection of intracellular cytokines as described above.

For adoptive transfer experiments, donor mice were injected with anti-ARTC2 nanobody (Treg-protector, Biolegend) 15 min before tissue harvesting. Single-cell suspension from spleen and thymus were prepared and red blood cells were lysed by incubation with lysis buffer. Next, cells were incubated with biotinylated anti-B220 antibody followed by Dynabeads biotin binder magnetic beads (Invitrogen) according to the manufacturer's instructions. Cells were resuspended in PBS and injected intravenously into congenic recipient mice. Tissues were harvested for analyses 12 days after transfer.

## RNA sequencing

RNA was extracted from sort-purified iNKT cells (TCRβ⁺CD1d-tet-PBS57⁺B220⁻CD11b⁻CD11c⁻CD8⁻) from the indicated tissues of WT mice using the RNAeasy micro kit (Qiagen) following manufacturer instructions. Library generation was performed according to manufacturer instructions using the

Nugen Ovation ultralow kit. Libraries were barcoded and run on an Illumina HiSeq 2500system with paired-end read lengths of 101 bp. Fastq files were trimmed using Cutadapt with a quality threshold of 10 before being aligned to and quantified against release GRCm38.p6 of the mouse genome with RSEM/Bowtie2. The raw counts were then imported into R/Bioconductor. DESeq2 was used to account for the different size factors between samples, and a model with main effects of tissue and sample-batch was used to find genes that were differentially expressed (false discovery threshold of 0.01) either between pairs of tissues (Wald test) or not constant across all tissues (likelihood ratio test). The RNAseq data are available in the Gene Expression Omnibus (GEO) database with accession number GSE131420.

### Analyses of TCR sequences

MiXCR software was used to identify TCR sequences within the RNAseq data (*Bolotin et al., 2013*). The software performs CDR3 extraction, identifies V, D and J segments, assembles clonotypes, filters out or rescues low-quality reads. The obtained repertoires were further filtered to eliminate out-of-frame and stop codon-containing CDR3 variants. CDR3 length, CDR3 physicochemical properties (hydrophobicity (based on the Kyte–Doolittle scale [*Kyte and Doolittle, 1982*]), isoelectric point (pI, according to EMBOSS [*Rice et al., 2000*]), frequency of polar (D, E, H, K, N, Q, R, S, T), aliphatic (A, I, L, V) or aromatic (F, H, W, Y) residues), CDR3 clonal size and V-J pairings were calculated using Brepertoire (*Margreitter et al., 2018*). CDR3α and CDR3β sequence logos were generated on the Weblogo server (*Crooks et al., 2004*) (https://weblogo.berkeley.edu) and provide a visual representation of amino acids enriched at different positions in the CDR3 sequences.

### Statistical analyses

Statistical analyses were performed using Prism software (GraphPad). Unless specified otherwise, *n* represents the number of individual mice analysed in each experiment. Statistical significance was determined using paired or unpaired two-tailed student's t test or ANOVA with multiple comparison Tukey test as specified in the figure legends. Correlation analyses were performed using Pearson correlation.

## Acknowledgements

This work was funded by the Medical Research Council (grant to PB MR/L008157/1); RJ was supported by a Marie Curie Intra-European Fellowship (H2020-MSCA-IF-2015–703639); ML-F is funded by a Francis Crick Institute-King's College London studentship; GA and BL are supported by a Medical Research Council Programme Grant (DKAA.RRAK18742). We thank the Science Technology Platforms from the Francis Crick Institute (which receives its core funding from Cancer Research UK, the UK Medical Research Council, and the Wellcome Trust). We acknowledge the NIH Tetramer Core Facility for provision of CD1d tetramers.

## Additional information

### Funding

| Funder | Grant reference number | Author |
|---|---|---|
| Medical Research Council | MR/L008157/1 | Patricia Barral |
| H2020 Marie Skłodowska-Curie Actions | Marie Curie Intraeuropean Fellowship (H2020-MSCA-IF-2015-703639) | Rebeca Jimeno |
| Medical Research Council | DKAA.RRAK18742 | Graham Anderson |

The funders had no role in study design, data collection and interpretation, or the decision to submit the work for publication.

## Author contributions
Rebeca Jimeno, Marta Lebrusant-Fernandez, Conceptualization, Data curation, Formal analysis, Investigation, Methodology; Christian Margreitter, Gavin Kelly, Franca Fraternali, Data curation, Formal analysis; Beth Lucas, Jo Spencer, Graham Anderson, Resources, Methodology, Investigation; Natacha Veerapen, Gurdyal S Besra, Resources; Patricia Barral, Conceptualization, Formal analysis, Supervision, Funding acquisition, Investigation, Methodology, Project administration

## Author ORCIDs
Franca Fraternali (iD) http://orcid.org/0000-0002-3143-6574
Patricia Barral (iD) https://orcid.org/0000-0003-4324-8973

## Ethics
Human subjects: Human tissues used in this study were collected with ethical approval from UK Research Ethics Committees administered through the Integrated Research Application System. All samples were collected with informed consent.
Animal experimentation: All animal experiments were approved by the Francis Crick Institute's and the King's College London's Animal Welfare and Ethical Review Body and the United Kingdom Home Office.

## Decision letter and Author response
Decision letter https://doi.org/10.7554/eLife.51663.sa1
Author response https://doi.org/10.7554/eLife.51663.sa2

# Additional files

## Supplementary files
• Transparent reporting form

## Data availability
The RNAseq data are available in the Gene Expression Omnibus (GEO) database with accession number GSE131420.

The following dataset was generated:

| Author(s) | Year | Dataset title | Dataset URL | Database and Identifier |
|---|---|---|---|---|
| Barral P, Jimeno R | 2019 | Analysis of transcriptomic profile of iNKT cells | https://www.ncbi.nlm.nih.gov/geo/query/acc.cgi?acc=GSE131420 | NCBI Gene Expression Omnibus, GSE131420 |

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
