## [Decision Letter]

**Acceptance summary:**

Tissue resident iNKT cells play an important role in regulating local immune responses. However, how tissue specific signals impact the properties of iNKT cells in different tissues remains unknown. Here, the authors show that iNKT cells in various tissues have distinct TCR repertoire, antigen specificity, activation status, and proliferative capacity. In addition, they show that TCR repertoire of iNKT cells is flexible as it is modulated after their arrival to peripheral organs and is further modified in responses to immunization and environmental changes. This study reveals that distinct TCR repertoire shaped by local environment may contribute to functional heterogeneity of iNKT cells in different anatomical sites. Thus, this work provides insights into tissue-specific immunoregulation mediated by iNKT cells.

**Decision letter after peer review:**

Thank you for submitting your article "Tissue-specific shaping of the TCR repertoire and antigen specificity of iNKT cells" for consideration by *eLife*. Your article has been reviewed by three peer reviewers, one of whom is a member of our Board of Reviewing Editors, and the evaluation has been overseen by Satyajit Rath as the Senior Editor. The following individual involved in review of your submission has agreed to reveal their identity: Thierry R Mallevaey (Reviewer #3).

The reviewers have discussed the reviews with one another and the Reviewing Editor has drafted this decision to help you prepare a revised submission.

Summary:

The manuscript by Jimeno et al. explored the diversity of the TCR repertoire of iNKT cells in distinct tissues. They found that the iNKT cell TCR repertoire differs between distinct tissues, and is different from recent thymic emigrants. There were also differences with regard to basal activation, proliferative capacity and clonal expansion. Additionally, the TCR repertoire was influenced by immunization, age and the environment. From these findings, the authors conclude that, after arrival in peripheral tissues, the iNKT cell population and its repertoire are further shaped.

Overall, this is a comprehensive and technically sound evaluation of the repertoire and phenotype of iNKT cells from mice and humans. The findings that TCR repertoire of iNKT cells is modulated after their arrival to peripheral organs and is modified in responses to immunization and environmental changes are quite novel. However, the differences in TCR repertoire are rather modest.

Essential revisions:

1) It would be interesting to study if/how the repertoire shifts in various tissues following the adoptive transfer of a known iNKT cell population.

2) The authors should comment on whether TCRVb usage affects the functionality of iNKT cells in response to antigen stimulation.

3) The authors should comment on whether TCR repertories is different among CD4^+^, DN, NK1.1+, and NK1.1- iNKT cell population in various organs.

4) It is somewhat surprising that the authors did not characterize TCR repertoire and antigen specificity of iNKT cells in the liver despite being the organ that is the most enriched for iNKT cells. The authors should at least perform FACS analysis and tetramer staining to address this issue.

5) What is the rationale to use OCH9 and not α-galactosylceramide or PBS57 for injections? Does α-galactosylceramide induce a similar shift in TCR repertoire?

6) The data appears inconsistent between Figure 2 and Figure 2—figure supplement 1 with respect to Vb8.1/8.2 usage. Have the authors phenotyped peripheral iNKT cells in Rag2-GFP mice using PD-1, CCR7 and Qa2 markers, to confirm the identity of the cells? How many GFP+ cells were acquired and analyzed in Figure 3? Please specify the age of the Rag2-GFP mice used in Figure 2.

---

## [Author Response]

Essential revisions:1) It would be interesting to study if/how the repertoire shifts in various tissues following the adoptive transfer of a known iNKT cell population.

We agree with the reviewers that this is an interesting question and we now provide this new data in Figure 1—figure supplement 1C of the revised manuscript. As suggested, we have adoptively transferred iNKT cells into congenic mice and analysed the TCRVb usage of cells found in various tissues of recipient animals 12 days after transfer. We detected the same tissue-dependent shift in TCRVβ usage in donor cells as that identified for endogenous cells, with a decreased frequency of Vβ8 and increased Vβ7 usage for iNKT cells found in the LNs in comparison with those found in the spleen.

2) The authors should comment on whether TCRVb usage affects the functionality of iNKT cells in response to antigen stimulation.

We have stimulated iNKT cells in vitro and in vivo with αGalCer and measured cytokine production by intracellular staining. Interestingly, we found that in both stimulatory conditions Vβ7^+^ iNKT cells produced significantly less IFN-γ and IL-4 than Vβ8^+^ or Vβother^+^ cells, suggesting that the TCRVβ usage relates to the functionality of iNKT cells in response to antigen stimulation. This new data has been included in the new Figure 6—figure supplement 2 of the revised manuscript.

3) The authors should comment on whether TCR repertories is different among CD4^+^, DN, NK1.1+, and NK1.1- iNKT cell population in various organs.

We have analysed the TCRVβ usage in iNKT cells from different tissues within the CD4^+^, NK1.1+, DP and DN subpopulations. We detected significant differences in TCRVβ usage both based on the iNKT cell subpopulations and on their tissue location. This new data complements and extends our previous findings and has been included in the new Figure 4—figure supplement 1 of the revised manuscript.

4) It is somewhat surprising that the authors did not characterize TCR repertoire and antigen specificity of iNKT cells in the liver despite being the organ that is the most enriched for iNKT cells. The authors should at least perform FACS analysis and tetramer staining to address this issue.

In this manuscript we specifically focused on iNKT cells from lymphoid tissues, hence we didn’t include characterisations for iNKT cells in the liver. We agree with the reviewers that this is an important question and we have performed flow cytometry analyses for iNKT cells from the liver as well as from other non-lymphoid tissues such as lung and small intestinal lamina propria. Our new data reveals differences in TCRVβ usage and lipid antigen binding amongst iNKT cells from non-lymphoid tissues. This new data has been included in Figure 1—figure supplement 1A-B and Figure 5—figure supplement 1 of the revised manuscript.

5) What is the rationale to use OCH9 and not α-galactosylceramide or PBS57 for injections? Does α-galactosylceramide induce a similar shift in TCR repertoire?

The use of OCH for immunisation experiments was based on the results obtained in Figure 5C and 5D, suggesting a preferential binding to OCH-loaded CD1d-tetramers by Vβ8^+^ iNKT cells. These results led us to hypothesize that OCH immunisation will induce the preferential expansion of Vβ8^+^ cells therefore causing a shift in the iNKT TCR repertoire which we indeed confirmed in Figure 6A-D.

As suggested, we performed immunisation experiments with αGalCer. We detected small but significant changes in the iNKT cell TCR repertoire 3 days after immunisation with an increase in the frequency of Vβ8^+^ cells at the expense of Vβother^+^ iNKT cells (the frequency of Vβ7^+^ iNKT cells remained unchanged). This new data is included in Figure 6—figure supplement 1 of the revised manuscript.

6) The data appears inconsistent between Figure 2 and Figure 2—figure supplement 1 with respect to Vb8.1/8.2 usage. Have the authors phenotyped peripheral iNKT cells in Rag2-GFP mice using PD-1, CCR7 and Qa2 markers, to confirm the identity of the cells? How many GFP+ cells were acquired and analyzed in Figure 3? Please specify the age of the Rag2-GFP mice used in Figure 2.

The differences in Vβ8 usage between NKT RTE from Rag2-GFP mice and PD1^-^CCR7^+^Qa2^low^ iNKT cells can be explained by the differences between these two populations. As suggested, we have stained GFP^+^ iNKT cells (from Rag2-GFP mice) for PD1, CCR7 and Qa2 and found that the GFP^+^ population doesn’t fully overlap with PD1^-^CCR7^+^Qa2^low^ cells. This data is in line with a previous report (Wang and Hogquist, 2018) that suggests that iNKT RTE are enriched for CCR7^+^ cells, but not all CCR7^+^ iNKT cells are RTE. Importantly, using both of these experimental approaches our data demonstrates that the repertoire of iNKT RTE is different from that of mature cells. We have included the new data in the new Figure 2—figure supplement 1B. The number of GFP+ cells ranged between 40 and 400 depending on the tissue. Rag2-GFP mice were 6-9-week-old. We have included these data in the revised manuscript and source data files.